# Half-Metallic Property Induced by Double Exchange Interaction in the Double Perovskite Bi_2_*BB′*O_6_ (*B, B′* = *3d* Transitional Metal) via First-Principles Calculations

**DOI:** 10.3390/ma12111844

**Published:** 2019-06-06

**Authors:** Hong-Zong Lin, Chia-Yang Hu, Po-Han Lee, Albert Zhong-Ze Yan, Wen-Fang Wu, Yang-Fang Chen, Yin-Kuo Wang

**Affiliations:** 1Institute Industrial Engineering of National Taiwan University, Taipei 10617, Taiwan; aj130820@gmail.com; 2Department of Applied Physics, College of Science, National Taiwan University, Taipei 10617, Taiwan; younger978@gmail.com (C.-Y.H.); yfchen@phys.ntu.edu.tw (Y.-F.C.); 3Department of Electro-Optical Engineering, National Taipei University of Technology, Taipei 10608, Taiwan; 4Affiliated Senior High School of National Taiwan Normal University, Taipei 10658, Taiwan; 5University of Southern California, Dornsife College of Letters, Arts, and Sciences, Los Angeles, CA 90089, USA; albertyanalbert@gmail.com; 6Department of Mechanical Engineering, National Taiwan University, Taipei 10617, Taiwan; wfwu@ntu.edu.tw; 7Center for General Education and Department of Physics, National Taiwan Normal University, Taipei 10610, Taiwan

**Keywords:** double perovskite, first-principle calculations, half-metal, double exchange, ferrimagnetic state

## Abstract

In this paper, we identify three possible candidate series of half-metals (HM) from Bi-based double perovskites Bi_2_*BB*′O_6_ (*BB*′ = transition metal ions) through calculations utilizing the density functional theory (DFT) and full-structural optimization, in which the generalized gradient approximation (GGA) and the strong correlation effect (GGA + *U*) are considered. After observing the candidate materials under four types of magnetic states, i.e., ferromagnetic (FM), ferrimagnetic (FiM), antiferromagnetic (AF), and nonmagnetic (NM), we found eight promising candidates for half-metallic materials. Under the GGA scheme, there are three ferromagnetic-half-metal (FM-HM) materials, Bi_2_CrCoO_6_, Bi_2_CrNiO_6_ and Bi_2_FeNiO_6_, and three FiM-HM materials, Bi_2_FeZnO_6_, Bi_2_CrZnO_6_ and Bi_2_CoZnO_6_. With implementation of the Coulomb interaction correction (GGA + *U*), we find two stable half-metallic materials: Bi_2_CrNiO_6_ and Bi_2_CrZnO_6_. We determine that the stability of some of these materials are tied to the double exchange interaction, an indirect interaction within the higher powers of localized spin interaction among transition metals via oxygen ions. Found in half-metallic materials, and especially those in the ferromagnetic (FM) state, the double exchange interaction is recognized in the FM-HM materials Bi_2_CrCoO_6_ and Bi_2_FeNiO_6_.

## 1. Introduction

Valued for their application in spintronic devices, half-metallic (HM) materials are useful based on their three distinguishing features: 100% spin polarization at the Fermi level (E_F_), quantization of magnetic moment, and zero spin susceptibility. Half-metals are characterized by an energy gap existing in one spin channel, showing insulating behavior, and no energy gap in the other spin channel, showing metallic behavior. This makes half-metallic materials particularly suitable for application in technologies such as computer memory manipulation, magnetic recording, single-spin electron source measurement, and high-efficiency magnetic sensors [1,2]. Some previously discovered half-metal materials include the doped perovskite structure manganese oxide La_0.7_Sr_0.3_MnO_3_ [1], spinel Fe_3_O_4_ [2], rutile CrO_2_ [3], double perovskites Sr_2_FeMoO_6_ [4,5,6,7,8,9], La_2_VTcO_6_ and La_2_VCuO_6_ [10], doped perovskite structure manganese oxide La_0.5_Ca_0.5_MnO_3_ [11], Mn-doping Mn_x_Ga_1−x_N [12], spinel FeCr_2_S_4_ [13], and Mn-doping GaAs [14,15]. 

In the search for new half-metallic materials, the double perovskite structure is a suitable template due to its flexibility and ease of illustrating an abundance of material properties, such as band insulators, ferromagnets, and ferroelectrics [3,16,17,18]. Organic double perovskites have recently been utilized for their remarkable optoelectronic properties [19,20,21,22,23,24,25]. Traditional photoelectrochemical cells, usually equipped with organic sensitizers and inorganic semiconductors, carry certain limitations that often result in low absorption coefficients and narrow absorption bands in these solar cells. However, seminal work by Kojima et al. in 2009 proposed a new material design based on organic-inorganic lead halide perovskite compounds CH_3_NH_3_PbBr_3_ and CH_3_NH_3_PbI_3_ in which a CH_3_NH_3_PbI_3_-based cell is able to convert solar energy with an efficiency of 3.8% and a CH_3_NH_3_PbBr_3_-based cell is able to gain a high photovoltage of 0.96 V [19]. Since this milestone, related research has been experiencing exponential growth to the point that photovoltaic efficiency has been able to reach above 22% [20]. There are also other designs of solar cells, such as efficient solar cells using iodide management in formamidinium-lead-halide–based perovskite layers, with a power conversion efficiency of 22.1% in small cells [22]. Furthermore, metal-halide perovskite semiconductors related to solar energy conversion have also become a hot topic [23,24]. Some Pb-free, stable inorganic double halide perovskites have been looked into as well since Pb is a pollutant element in our environment [25].

However, our studies focus on inorganic double perovskites, such as the Sr-based double perovskites Sr_2_*BB*′O_6_, including Sr_2_FeMoO_6_ [4,5,6,7,8,9], Sr_2_FeReO_6_ [5,26], Sr_2_FeWO_6_ [6], Sr_2_CrMoO_6_ [7], Sr_2_CrReO_6_ [26,27], and Sr_2_CrWO_6_ [5,28], all of which are HM materials. We use an *A*_2_*BB*′O_6_ structure constructed by two different ternary perovskites, *AB*O_3_ and *AB*′O_3_, where the *A*-site elements are substituted with alkaline earth or rare-earth ions, such as Ca, Sr, Ba, and La, and the *B*-site elements are substituted with transition metal ions, utilized for their diverse electronic configurations in d-orbitals and juxtaposition due to the strong interaction. Dissimilarities in size and valence between the *B* and *B′* ions are vital in controlling the physical properties of these materials [29,30]. By exploring a multitude of compounds based on this structure, we can easily identify suitable HM candidates.

## 2. Materials and Methods

This paper mainly focuses on the search for HM materials from inorganic double perovskite compounds. Accordingly, we expand on previous outcomes and execute calculations in the area of Bi-based double perovskites Bi_2_*BB*′O_6_ (*B, B′* = *3d* transition metal) with consideration of 4 types of initial magnetic states, namely ferromagnetic (FM), ferrimagnetic (FiM), antiferromagnetic (AF), and nonmagnetic (NM), with a total of 45 ($C_{2}^{10}$) compounds of Bi_2_*BB′*O_6_ series in these studies. Additionally, we compare the AF calculations of these materials with their FM-based calculations. After the process of full structural optimization, we attempt to find out how many cases of stable HM materials are left and analyze the *d*-orbital distributions of the density of states (DOS) in these materials. We go a step further and determine the cases of HM materials in view of the strong correlation correction (GGA + *U*) as well.

We also investigate the effect of the double exchange (DE) interaction within these HM candidates. The first proposal of DE was presented by Zener [31], who considered the DE to involve simultaneous jumps between two electrons. A second description was proposed by Anderson and Hasegawa [32] who considered DE to be the transfer process in which a mobile electron is able to move one Mn ion to the other Mn ion by the so-called local exchange. Some theoretic and calculation works related to the DE effect are described in detail in previous studies [8,9,17,33,34,35,36].

### 2.1. Structural Optimization

To utilize and calculate an atomic model, it is often necessary to have structural optimization to ensure accuracy. In order for the ideal cubic structure (space group of $Fm\bar{3}m$, No. 225) to be assigned as the F(i)M or AF state, we have to organize the *B* and *B*′ ions in the order of a NaCl configuration with a lattice constant of 2*a*. According to this ordering, the *B*(*B*′) elements are equipped with transition metals with an O ion placed evenly between each pair. For an ideal cubic structure, the *c*/*a* ratio is set to $\sqrt{2}$. To understand the observed stable states in the condition of a supercell (2 formula units [f.u.]), we relax this cubic structure to a reduced symmetry through structural optimization calculations. 

Next, we determine whether the structures are stable after full structural optimization, resulting in two structures being discussed. One of them is the tetragonal structure (space group of *I4/mmm*, No. 139) accompanied by two non-equivalent types of O atoms, in which two O_1_ atoms are located on the *z*-axis and four O_2_ atoms are located on the *xy*-plane, as shown in Figure 1. Accordingly, these are the cases of F(i)M state. The other one is the tetragonal structure (space group of *P4/mmm*, No. 123) with non-equivalent types of O atoms, in which the angle of *B*-O_1_-*B*′ remains at 180° and the angle of *B*-O_2_-*B*′ has been altered a little but is still close to 180° during structural optimization. This means the symmetry reduction is deemed rather minor and the *c*/*a* ratio is very close to the value of $\sqrt{2}$. This final structure is found in the AF state. 

### 2.2. Calculation Method and Procedure

For the initial magnetic configuration of the *B* ion pairs and *B*′ ion pairs, we assign the magnetic states for the initial stage based on the four states of FM, FiM, AF, and NM, as shown in Figure 2. The spin alignment of FM state is assigned as (*B*, *B*, *B*′, *B*′) = (*m*, *m*, *m*′, *m*′), the spin alignment of FiM state is assigned as (*B*, *B*, *B*′, *B*′) = (*m*, *m*, −*m*′, −*m*′), and AF as (*B*, *B*, *B*′, *B*′) = (*m*, −*m*, *m*′, −*m*′). For NM, there is no spin assigned initially. All the cases begin with the ideal cubic structure belonging to the space group $Fm\bar{3}m$.

By going through the process of full structural optimization in the GGA scheme, some cases for FM and FiM states converge into the other state, while others remain in their initial states. Furthermore, we assign the magnetic case as half-metallic anti-ferromagnetic (HM-AF) if the summation of the total magnetic moment with an initial FiM state becomes zero [10] and the induced symmetry of the spin-up and spin-down orbitals exists in the total density of state (DOS). The cases related to the calculations of spin polarization have always been more stable than those of the NM state.

The calculations are based on Density Functional Theory (DFT) [37] and the full-potential projector augmented wave method [38], which is implemented in the Vienna Ab initio Simulation Package (VASP) code [39,40,41]. In searching for stable structures and stable ionic positions, we choose the full structural optimization process to relax both lattice constants and atomic positions, while applying the conjugate-gradient method to relax the ionic positions. The related parameters are set as follows: the energy convergence criteria for self-consistent calculations is set as 10^−6^ eV, 8 × 8 × 6 *k*-point grids are used for the Brillouin zone, the cut-off energy of the plane wave basis is set as 450 eV, the Wigner–Seitz radius of the Sr atom set as 2.5 atomic units (a.u.), 1.4 a.u. for O atom, and 2.1 a.u. as *B*(*B*′) ion. For the final and equilibrium structures, the forces and stresses acting on all the atoms were less than 0.03 eV/Å and 0.9 kBar, respectively. 

After the structural optimization, the F(i)M generally reduces to a tetragonal structure (space group of *I4/mmm*, No. 139), while the AF state will be in a different tetragonal structure (space group of *P4/mmm*, No. 123). The NM state is always unstable in consideration of the energy difference between the other states. Hence, we will exclude the discussion of NM cases in the following sections. Another condition that will be taken into account is the electron correlation correction (GGA + *U*), which will be applied for all the cases. We choose the effective parameter *U_eff_*, *U − J*, where *U* is defined as the Coulomb potential and *J* as the exchange parameters. 

In this paper, the parameter *U_eff_* is denoted as *U* for simplicity in the functions of the *d* orbital. Dealing with the transition metals, the nearly maximum values are selected from a reasonable range of *U* [42]. In this study, consequently, *U* values were assigned as 2.0, 3.0, 4.0, 5.0, 6.0, 6.0, 7.0, and 7.0 for Sc, Cr, Mn, Fe, Co, Ni, Cu, and Zn, respectively. Near maximum is chosen because when considering two cases of initial *U* = 0 and less than 0.5 ×
*U*, the calculation results with less than 0.5 ×
*U* are almost the same in comparison with the case of *U* = 0. Therefore, selecting the nearly maximum values of *U* is needed when trying to disclose the results from strong Coulomb interaction in these compounds.

## 3. Results and Discussion

After the process of full structural optimization, 8 of the 45 compounds in the Bi_2_*BB*′O_6_ series were categorized as HM materials. These include 6 compounds in the GGA scheme, namely Bi_2_CrCoO_6_, Bi_2_CrNiO_6_, and Bi_2_FeNiO_6_ as FM-HM materials, and Bi_2_FeZnO_6_, Bi_2_CrZnO_6_ and Bi_2_CoZnO_6_ as FiM-HM materials. With the scheme of (GGA + *U*), 2 stable half-metallic (HM) materials Bi_2_CrNiO_6_ (FM-HM) and Bi_2_CrZnO_6_ (FiM-HM) are found. Figure 3, Figure 4, Figure 5, Figure 6, Figure 7 and Figure 8 describe the DOS and the partial DOS (PDOS) for these 6 FM/FiM-HM stable materials in the order of Bi_2_CrCoO_6_, Bi_2_CrNiO_6_, Bi_2_FeNiO_6_ Bi_2_FeZnO_6_, Bi_2_CrZnO_6_, and Bi_2_CoZnO_6_, respectively. Figure 9 and Figure 10 illustrate the 2 AF-based compounds, Bi_2_CrCoO_6_ and Bi_2_FeNiO_6_. For a clearer illustration of each Bi_2_*BB*′O_6_, every sub-Figure a represents the total DOS and sub-Figure b,c demonstrate the PDOS for *B* and *B′* atoms with the GGA scheme. Sub-Figure d-f represents the total DOS with the GGA + *U* scheme, and the PDOS for *B* and *B′* with the GGA + *U* scheme, respectively. To carefully monitor the strong-correlation correction (GGA + *U*) for transition metals, the *U* values are noted as (*U_B_*, *U_B_*_′_). After optimization, FM and FiM states converge, with structural parameters of their tetragonal structure (*I4/mmm*, No. 139) illustrated in Table 1. AF retains its state with the tetragonal structure (P4/mmm, No. 123). After fully optimizing Bi_2_*BB*′O_6_, two types of oxygen are labeled as O_1_ and O_2_, of which the final stable positions of O_1z_, O_2x_ and O_2y_ are listed in Table 1. The volume variations are slight for all the compounds, except for Bi_2_CrZnO_6_, which has the largest volume of 118.095 Å^3^/f.u. In addition, the calculated physical properties and energy differences are listed in Table 2, where we are able to find energy differences, △E, between the FM and AF states. Table 3 indicates all the energy states for various final magnetic states in detail. The energy levels for the final NM states are all larger than those of AF and FM(FiM) by roughly 1000 meV or above, implying that the NM cases are unstable. This is why we have excluded the list of NM energy states here.

### 3.1. FM-HM Compounds: Bi_2_CrCoO_6_, Bi_2_CrNiO_6_, and Bi_2_FeNiO_6_

Here, Bi_2_CrCoO_6_, Bi_2_CrNiO_6_, and Bi_2_FeNiO_6_ lean towards the FM state after structural optimization, of which the calculated total energy differences (△E = FM − AF) are −25, −97, and −43 meV/f.u., respectively. These 3 cases are all stable FM-HM compounds in the GGA scheme. However, with the GGA + *U* scheme, the △E are 903, −112, and 34 meV/f.u., respectively. Only Bi_2_CrNiO_6_ remains a stable FM-HM compound for both GGA and GGA + *U* schemes. The data are listed in Table 2 and Table 3.

The band gaps occurs in the spin-down channel for the compounds Bi_2_CrCoO_6_ and Bi_2_CrNiO_6_, as shown in Figure 3a and Figure 4a, while the band gap exists in the spin-up channel for Bi_2_FeNiO_6_ as shown in Figure 5a. Monitoring the location of the band gap near Fermi energy provides the most direct evidence for HM material. The second indicator for HM material is related to the integer value of total magnetic moments (*m*_tot_), where *m*_tot_ = 3.0 *μ*_B_/f.u. is noted for Bi_2_CrCoO_6_ (GGA), 4.0 *μ*_B_/f.u. for Bi_2_CrNiO_6_ (GGA and GGA + *U*), and Bi_2_FeNiO_6_ (GGA). However, there were two cases that lost the possibility of being HM material when the GGA + *U* scheme was considered. The first one was because the total magnetic moment of Bi_2_CrCoO_6_ changed from 3.0 to 6.792 *μ*_B_/f.u.; the second case was when Bi_2_FeNiO_6_ lost its magnetic stability for FM when considering the energy of its AF state. 

In these compounds, the *p* orbitals of Oxygen have two kinds of energy distributions for DOS. For example, in Bi_2_CrCoO_6_, the hybridization between Cr *t_2g_*, O 2*p*, and Co *t_2g_* orbitals occurs mainly in the two energy ranges −7.5 eV to −2.5 eV and −2.0 eV to 0.5 eV, as illustrated in Figure 3a. The localized spin-to-spin interaction of double exchange was inferred from the interaction of Cr*_t2g_*-O*_2p_*-Co*_t2g_*. Such DE effect enhances the HM property of Bi_2_CrCoO_6_. Likewise, in Bi_2_CrCoO_6_, a band gap exists at the spin-down channel with the GGA scheme, but the gap disappears and its FM state becomes unstable once GGA + *U* is applied, thus losing its half-metal properties.

In the GGA scheme, the weak magnetic moment 0.162 *μ*_B_ for Co is caused by the asymmetrical distribution of the Co *e_g_* near the Fermi level as shown in Figure 3b,c, while in the GGA + *U* scheme, the stronger and non-integer magnetic moment of 3.071 *μ*_B_ for Co is induced again from the variation of the *t_2g_* orbital of Co. According to the energy listed in Table 2, there is a huge variation for Bi_2_CrCoO_6_ from GGA to GGA+U scheme, −25 to 903 meV. For such a result, we have to take a deeper look into the PDOS of the Cr and Co atoms. When applying the GGA + *U* scheme for Bi_2_CrCoO_6_, the PDOS of Cr *e_g_* near Fermi level is pushed to a higher energy level, while the Cr *t_2g_* only changes a little. Furthermore, the PDOS of Co *t_2g_* (spin-up) shifts back to a lower energy level, while the PDOS of Co *t_2g_* (spin-down) is pushed to a higher energy level, crossing the Fermi energy and breaking the gap. There is a huge variation within the PDOS of Co *e_g_*. All of these variations of PDOS result in the total energy value changing from negative to positive. Consequently, AF becomes the stable state, as shown in Table 3. The energy gap disappearing in the DOS is due to the variation of Cr and Co atoms, as shown in Figure 3d.

Next, we investigate the valence configurations. Using the nominal valence states, the ordered double perovskite points to the state of Bi_2_^3+^(*BB*′)^6+^O_6_^2−^. According to the magnitude and spin direction of the Cr/Co magnetic moment, the transition metal atoms *B* and *B**′* of Bi_2_CrCoO_6_ can have valence configurations of Cr^+3^(3*d*^3^) and Co^+3^(3*d*^6^), so that Cr ^3+^(*3d*^3^*4s*^0^:*t^3^_2g_e^0^_g_*) at S = 3/2 and Co^3+^(*3d*^6^*4s*^0^:*t^6^_2g_e^0^_g_*) at S = 0. Moreover, in the actual materials, the electronic number from the mentioned ionic model, with simplified valence configurations, would be redistributed in consideration of the process of hybridization among Cr (Co) 3*d* and O 2*p* orbitals. This results in Cr and Co having total election numbers of 4.4 and 7.3 for the *d* orbitals, implying the valence states of Cr^+1.6^(3*d*^4.4^), Co^+1.7^(3*d*^7.3^) for Bi_2_CrCoO_6_. For GGA + *U*, the valence states of Cr^+1.6^(3*d*^4.4^) and Co^+1.8^(3*d*^7.2^) are almost the same. Notably, the gap disappears for the GGA + *U* scheme.

For Bi_2_CrNiO_6_, the solid evidence of being half-metallic arises from the integer total magnetic moment of 4.0 *μ*_B_ and spin-down energy gap of 0.65 (1.73) eV for both GGA and GGA+*U* processes. The process of GGA+*U* does not destroy the energy gap because the *e_g_* and *t_2g_* of Ni drop to a lower energy level, while the spin up channel maintains the conduction band. As a result, a stable FM-HM Bi_2_CrNiO_6_ is determined. The electron configuration for the ideal ionic model is Cr^+3^ (*3d*^3^*4s*^0^:*t^3^_2g_e^0^_g_*) at S = 3/2 and Ni^+3^(3d^7^4s^0^: *t^6^_2g_e^1^_g_*) at S = 1/2. According to the data listed in Table 2, the calculated electron numbers of *d* orbitals for Cr and Ni are 4.4 and 8.3, giving the valence states of Cr ^+1.6^(3*d*^4.4^) and Ni^+1.7^(3*d*^8.3^). In consideration of the GGA + *U* scheme, similar to the case with GGA, Cr and Ni have *d* orbital electron numbers of 4.3 and 8.3 and valence states of Cr^+1.7^(3*d*^4.3^) and Ni ^+1.7^(3*d*^8.3^). 

Following that, we explore the compound Bi_2_FeNiO_6_, in which the evidence of half-metallic property exists from the integer total magnetic moment of 4.0(6.0) *μ*_B_ and energy gaps of 0.15 (1.68) eV in both GGA and GGA + *U* processes. In the case of the GGA + *U* scheme, however, the stability of the FM state disappears and AF becomes the final stable state. The gap near the Fermi level also flips and enlarges. This ensues from the significant *d* orbital variations of the PDOS for Fe and Ni between the FM and AF states. 

The orbital hybridization among Fe *t_2g_*, O *2p*, and Ni *t_2g_* occurs mainly in the two parts of energy distribution from −7.5 eV to −0.8 eV and −0.2 eV to 1.0 eV, as shown in Figure 5a. The localized spin-to-spin interaction double exchange that enhances the HM property originates from the interaction of Fe*_t2g_*-O*_2p_*-Ni*_t2g_*. However, when the case with GGA + *U* scheme is taken into consideration, the fact that the energy gap near the Fermi level flips from spin-up to spin-down and the *d* orbital distributions of Fe and Ni change largely results in the instability of the FM state. For the ideal ionic model of this compound, the electron configuration is Fe^+3^ (*3d*^5^4s^0^:*t^5^_2g_e^0^_g_*) at S = 1/2 and Ni^+3^ (3d^7^4s^0^:*t^6^_2g_e^1^_g_*) at S = 1/2. Consequently, after the calculation for the electron distribution of *d* orbitals of Bi_2_FeNiO_6_, the electron numbers for the *d* orbitals of Fe and Ni are 6.2 and 8.3. This result provides the real valence states of Fe^+1.8^(3*d*^6.2^) and Ni^+1.7^(3*d*^8.3^). When considering the GGA *+ U* scheme, the valence states are calculated as Fe^+2.0^(3*d*^6.0^) and Ni^+1.7^(3*d*^8.3^). Only the calculation of Bi_2_FeNiO_6_ under the GGA scheme provides a stable state of FM-HM. For Bi_2_FeNiO_6_ under the GGA *+ U* scheme, it becomes unstable as FM due to the energy gap flip and the changes in the *d* orbitals of Fe and Ni. To recap, we found stable FM-HM cases in Bi_2_CrCoO_6_ (GGA), Bi_2_CrNiO_6_ (GGA and GGA *+ U*), as well as Bi_2_FeNiO_6_ (GGA).

### 3.2. FiM-HM Compounds: Bi_2_FeZnO_6_, Bi_2_CrZnO_6_ and Bi_2_CoZnO_6_

To determine the final stable state for Bi_2_FeZnO_6_, Bi_2_CrZnO_6_, and Bi_2_CoZnO_6_, we execute the calculation of structural optimization, in which all FM (FiM) states converge to either FiM or FM states. Consequently, in the GGA scheme, the calculated total energy differences (△E = FM − AF) are listed as −58, −60, and −51 meV/f.u. for Bi_2_FeZnO_6_, Bi_2_CrZnO_6_, and Bi_2_CoZnO_6_, respectively, while with the GGA + *U* scheme, △E are recorded as 1125, −73, 99 meV/f.u. Furthermore, in the GGA scheme, energy gaps near Fermi level occur at the spin-down channel for Bi_2_CrZnO_6_ and at the spin-up channel for Bi_2_FeZnO_6_ and Bi_2_CoZnO_6_. The energy gaps, in GGA, are reported as 0.97, 1.30, 0.85 eV, while with the GGA + *U* scheme, the energy gaps are 1.52 and 1.57 eV for Bi_2_FeZnO_6_ and Bi_2_CrZnO_6_. There are two energy gaps very close to Fermi level in Bi_2_CoZnO_6_ for spin-up and spin-down when applying the GGA + *U* scheme. We observe integer magnetic moments of 2.0 *μ*_B_/ f.u. for Bi_2_CrZnO_6_ in both GGA and GGA + *U* schemes. On the other hand, the magnetic moments changed from 2.0 to 4.0 *μ*_B_/f.u. and 1.0 to 5.0 *μ*_B_/f.u. for Bi_2_FeZnO_6_ and Bi_2_CoZnO_6_ after the GGA + *U* scheme was given consideration. All the data are listed in Table 2 and Table 3.

According to the PDOS of Bi_2_FeZnO_6_, the hybridization of Fe *t_2g_* and O 2p occur in the range of energy from −7.5 eV to −1.0 eV for the spin-up channel and −0.5 eV to 0.5 eV for the spin-down channel. One weak magnetic moment −0.022 *μ*_B_ for Zn is induced by the asymmetry of the Zn *e_g_* at the Fermi level. Hybridization between the Zn 3*d* and O 2*p* orbitals from −7.5 eV to −2.0 eV and −0.8 eV to 0.5 eV in the spin down channel is regarded as the main reason for the FiM state, as shown in Figure 6. The total electron numbers of *d* orbitals for Fe and Zn are 6.2 and 9.9, respectively. This gives the valence states of Fe^+1.8^(3*d*^6.2^) and Zn^+2.1^(3*d*^9.9^). When considering the ideal ionic model, the electron configuration is Fe^4+^(*3d*^4^*4s*^0^:*t^4^_2g_e^0^_g_*) at S = 1, Zn^2+^(*3d*^10^*4s*^0^:*t^6^_2g_e^4^_g_*) at S = 0. The electron configurations change a little after GGA + *U*, with Fe^+1.9^(3*d*^6.1^) and Zn^+1.9^(3*d*^10.1^). Not only does the band gap switch from a spin-up to a spin-down after GGA + *U*, but the △E is also extremely high. The energy instability has removed Bi_2_FeZnO_6_ (GGA + *U*) from candidacy as a stable FiM-HM.

Figure 7 illustrates the density of states (DOS) and PDOS of Bi_2_CrZnO_6_ for both GGA and GGA + *U* (Cr = 5, Zn = 7) schemes. With the GGA scheme, the hybridization between the Cr 3*d* and O 2*p* orbitals occurs mainly in the energy regions −7.0 eV to −1.0 eV and −0.5 eV to 0.5 eV, as shown in Figure 7a. Principally near the Fermi level (E_F_), a spin-splitting at the spin-down channel and integer magnetic moment of 2.0 *μ*_B_/f.u. is regarded as the main features of HM. This asymmetry of the Zn *e_g_* at the Fermi level gives Zn a weak magnetic moment, induced as −0.013 (GGA) and −0.008 (GGA + *U*) *μ*_B_/f.u. These factors result in the FiM state. Next, we investigate the distribution of the electron configuration, for which the ideal model of Bi_2_CrZnO_6_ is Cr^4+^(*3d*^2^*4s*^0^: *t^2^_2g_e^0^_g_*) at S = 1, Zn^2+^(*3d*^10^: *t^6^_2g_e^4^_g_*) at S = 0. According to the following calculations, the total electron numbers of *d* orbitals for Cr and Zn are 4.4 and 10.0, respectively. This gives the valence states of Cr^+1.6^(3*d*^4.4^) and Zn^+2.0^(3*d*^10.0^). Considering the electron configuration with GGA + U scheme, it is also similar to the configuration with GGA. After GGA + *U*, the spin-down band gap slightly widens due to a shift in the Cr *t_2g_* orbital and △E remains negative, showing a stable FiM-HM state. Similar to Bi_2_FeZnO_6_, illustrated in Figure 6, Bi_2_CrZnO_6_ is regarded as a stable FiM-HM for both GGA and GGA+*U* schemes. 

Finally, in the compound Bi_2_CoZnO_6_, the hybridization between Co *t_2g_* and O 2*p* occurs in the two energy regions −7.0 eV to −0.6 eV at the spin-up channel and −1.0 eV to 0.4 eV at the spin-down channel, as shown in Figure 8a. There is a weak magnetic moment of −0.013 *μ*_B_ for the Zn atom due to the asymmetric distribution of Zn *e_g_* at the Fermi level. The energy gap of 0.85 eV exists at the spin-up channel with GGA, while with the GGA *+ U* scheme, there are energy gaps of 0.65 eV for the spin-up channel and 0.75 eV for the spin-down channel, resulting in an insulating material and not a half-metal. Considering the electron configuration, the suitable ionic model for Bi_2_CoZnO_6_ is Co^4+^(*3d*^5^*4s*^0^:*t^5^_2g_ e_g_*) at S = 1/2, Zn^+2^(*3d*^10^:*t^6^_2g_e**^4^_g_*) at S = 0. After calculations, the total electron numbers of *d* orbitals for Co and Zn are estimated to be 7.2 and 9.9. With GGA, this results in the final valence states of Co^+1.7^(3*d*^7.3^) and Zn^+2.1^(3*d*^9.9^). The electron configurations are regarded as Co^+2.0^(3*d*^7.0^) and Zn^+2.0^(3*d*^10.0^) if GGA*+U* is considered. Consequently, Bi_2_CoZnO_6_ (GGA) is determined to be a stable FiM-HM material.

### 3.3. AF Compounds: Bi_2_CrCoO_6_ and Bi_2_FeNiO_6_

In this section, we select two compounds, Bi_2_CrCoO_6_ and Bi_2_FeNiO_6_, as examples to discuss AF-based calculations because of their final stable states, with GGA + *U* schemes belonging to AF rather than FM. We illustrate the distribution of DOS and PDOS of Bi_2_CrCoO_6_ and Bi_2_FeNiO_6_ in Figure 9 and Figure 10, respectively. Comparing the states of Bi_2_CrCoO_6_ in Figure 3 with Figure 9, the GGA+*U* scheme has changed the compound from a FM-HM to a stable AF-Insulator (AF-IS). Meanwhile, the GGA + *U* scheme changed Bi_2_FeNiO_6_ from a FM-HM to a stable AF-Metal as shown in Figure 5 and Figure 10. When considering the AF-based states, the total magnetic moment is zero because the magnetic alignments of B and B′ are anti-parallel, as shown in Table 2. Thus, this table only lists the magnetic moment for one site of *B* and *B′* and omits another site. In addition, for analysis of stability, the energy level of AF-IS Bi_2_CrCoO_6_ (GGA + *U*) is −61.827 eV/f.u. and AF-metal Bi_2_FeNiO_6_ (GGA + *U*) is −57.083 eV/f.u., shown in Table 3. 

In view of the total electron numbers for the *d* orbitals in AF Bi_2_CrCoO_6_, Cr and Co have electron numbers of 4.4 and 7.3, implying the valence states of Cr ^+1.6^(3*d*^4.4^), Co^+1.7^(3*d*^7.3^) for GGA. For GGA + *U*, the valence states are Cr^+1.6^(3*d*^4.4^) and Co^+1.9^(3*d*^7.1^). The PDOS distribution of AF Bi_2_CrCoO_6_ is similar to that of FM Bi_2_CrCoO_6_. Considering the distribution of AF Bi_2_FeNiO_6_, these results provide the valence states of Fe^+1.9^(3*d*^6.1^) and Ni^+1.7^(3*d*^8.3^) for GGA and Fe^+2.0^(3*d*^6.0^) and Ni^+1.7^(3*d*^8.3^) for GGA *+ U*. All the AF data are listed in Table 2 and Table 3. Finally, the stable states under GGA *+ U* are AF-IS Bi_2_CrCoO_6_ and AF-metal Bi_2_FeNiO_6_. However, these two compounds are not HM material.

### 3.4. Double Exchange Interaction

The evidence of double exchange (DE), favoring ferromagnetism, is investigated in this section. Double exchange is related to the indirect exchange process constructed by two magnetic ions (*BB**′*) with localized spin Si and Sj via local exchange. Such an exchange minimizes the kinetic energy of hopping electrons due to the alignment of neighboring magnetic ions (*BB**′*). Hence, it typically exists in compounds with mixed valence magnetic ions, especially in the transition metal (TM) ions. There are two factors included here for DE: one is the localized spin at each TM ion site and the other is Hund’s first rule, which shows that the localized spin will couple with the spin of the mobile electron. Such an effect is interpreted as the related characteristic scalar product (S*i*·S*j*). The derivation of DE Hamiltonian in detail can be referred to in a previous study [34]. 

Figure 11a,b demonstrates a clear double exchange interaction between the *BB*′, Bi_2_CrCoO_6_ and Bi_2_FeNiO_6_. In the GGA scheme, for *B* = Cr and *B**′* = Co, the spin-up electrons transfer from occupied Co *d* to empty Cr *d* in the *t_2_g* (spin-up state) via O 2p, accompanying the FM-based material. Notably, for Bi_2_CrCoO_6_, there is distinct double exchange among the interaction of Cr*_t2g_*-O*_2p_*-Co*_t2g_*, while for Bi_2_FeNiO_6_, the double exchange originates from the interaction of Fe*_t2g_*-O*_2p_*-Ni*_t2g_*. 

Due to the fact that double exchange is a kind of mechanism happening between two magnetically active ions, i.e., partially-filled d-shells with different charge statuses, double exchange is irrelevant here in compounds with Zn fully-filled d-orbitals. Consequently, it is not possible for the electrons from Co to hop to the Zn site (see the PDOS of Zn in Figure 7) so no double exchange exists here. Accordingly, the DE effect is not discussed for the cases of Bi_2_FeZnO_6_, Bi_2_CrZnO_6_, and Bi_2_CoZnO_6_. According to the electronic configurations mentioned in the discussion, we found evidence of double exchange existing in Bi_2_CrCoO_6_ and Bi_2_FeNiO_6_ FM-HM materials. The double exchange interaction of *B*(*t*_2g_)-O(2*p*)-*B**′*(*t*_2g_) configuration enhances the HM properties for these compounds and provides reasonable explanations for the stable magnetic states, magnetic moments, and energy gaps occurring in the spin channels.

## 4. Conclusions

This work has systematically investigated the electronic structure and magnetic properties of the double perovskite oxides Bi_2_*BB*′O_6_ (*B, B′* as *3d* transitional metal). Accompanied by density functional theory (DFT) and full-structure optimization by generalized gradient approximation (GGA) and the strong correlation effect (GGA + *U*), a thorough examination of the possibility of HM materials under the four types of initial magnetic states was completed, i.e., ferromagnetic (FM), ferrimagnetic (FiM), antiferromagnetic (AF), and nonmagnetic (NM). The results indicate that there are six possible stable FM/FiM-HM materials, containing three FM-HM materials—Bi_2_CrCoO_6_, Bi_2_CrNiO_6_, and Bi_2_FeNiO_6_—and three FiM-HM materials—Bi_2_FeZnO_6_, Bi_2_CrZnO_6_ and Bi_2_CoZnO_6_. When the Coulomb interaction correction (GGA + *U*) is considered, there are two promising candidates for half-metallic (HM) materials, which are Bi_2_CrNiO_6_ and Bi_2_CrZnO_6_. The evidence of double exchange interaction is disclosed from the Bi_2_CrCoO_6_ and Bi_2_FeNiO_6_ by the hybridization of magnetic ions’ 3*d* orbitals via O 2*p*. Moreover, we also explain why some calculated stable AF states are not also suitable candidates for HM material. Hopefully, through this research process using inorganic double perovskite oxides, we are able to provide a solid pathway for surveys of possible HM candidates and encourage scientists to execute further experimental research studies on these related HM materials.

## Figures and Tables

**Figure 1 materials-12-01844-f001:**
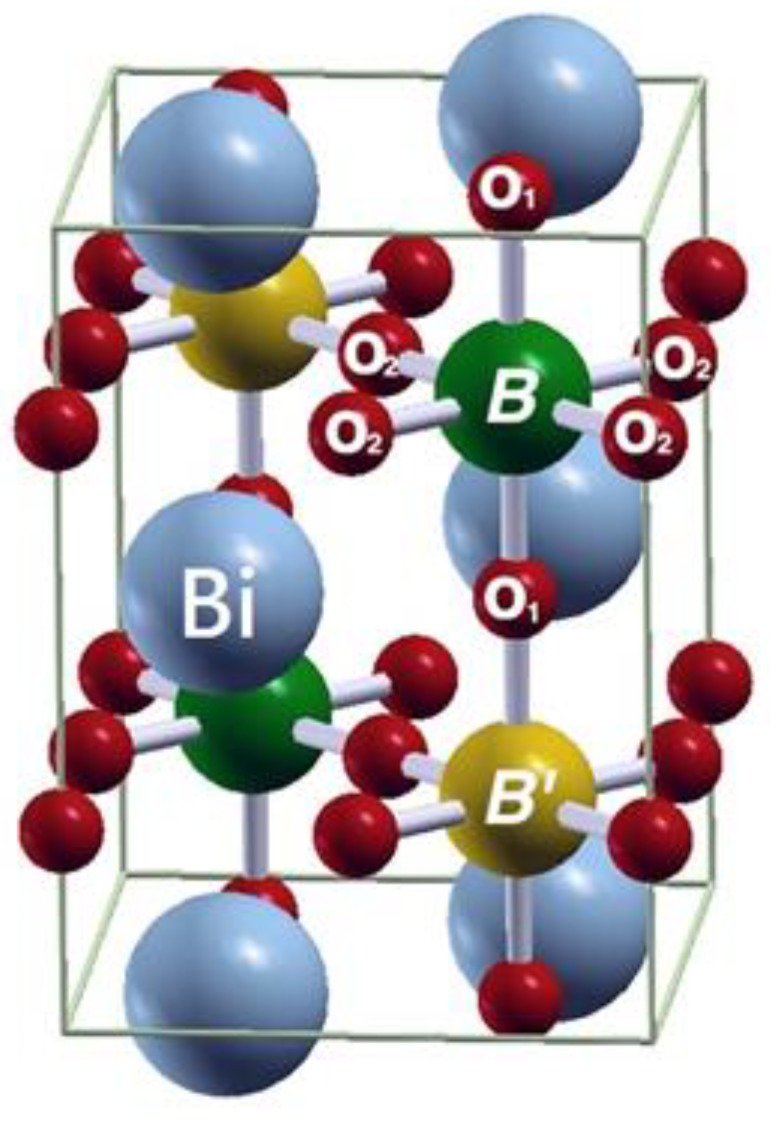
An ideal ordered double perovskites structure Bi_2_*BB*′O_6_. (*B, B′* = *3d* transition metal).

**Figure 2 materials-12-01844-f002:**
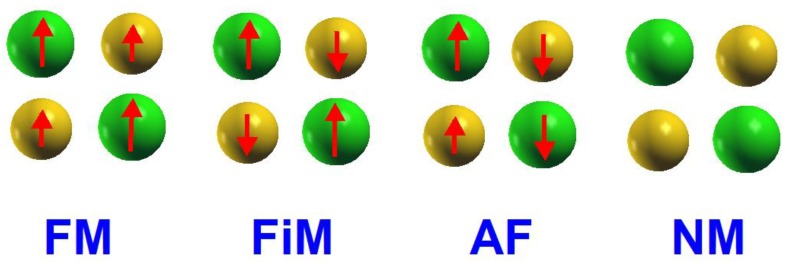
The schematic diagram of 4 magnetic states: FM, FiM, AF, and NM.

**Figure 3 materials-12-01844-f003:**
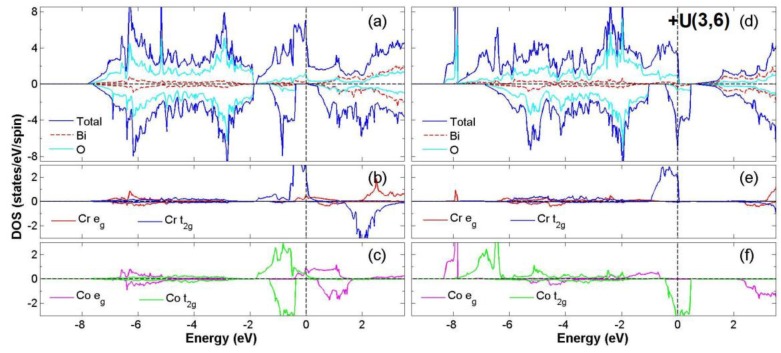
Calculated FM-Bi_2_CrCoO_6_ total Density of State (DOS) (**a**) and partial DOS of *e_g_* and *t_2g_* spin orbitals for Cr (**b**) and Co (**c**) under GGA and total DOS (**d**) partial DOS of *e_g_* and *t_2g_* spin orbitals for Cr (**e**) and Co(**f**) under GGA + *U* (Cr = 3, Co = 6) schemes.

**Figure 4 materials-12-01844-f004:**
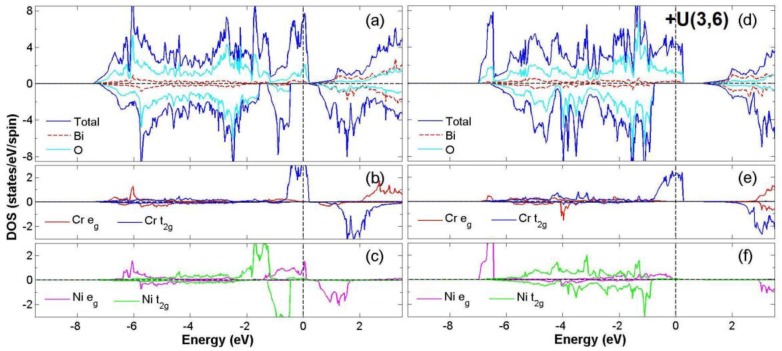
Calculated FM-Bi_2_CrNiO_6_ total DOS (**a**) and partial DOS of *e_g_* and *t_2g_* spin orbitals for Cr (**b**) and Ni (**c**) under GGA and total DOS (**d**) partial DOS of *e_g_* and *t_2g_* spin orbitals for Cr (**e**) and Ni (**f**) under GGA + *U* (Cr = 3, Ni = 6) schemes.

**Figure 5 materials-12-01844-f005:**
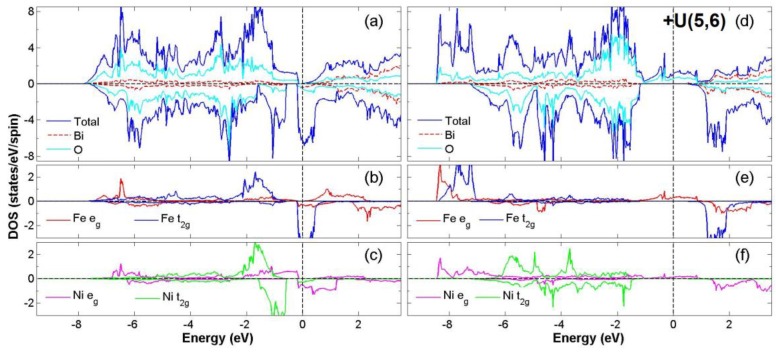
Calculated FM-Bi_2_FeNiO_6_ total DOS (**a**) and partial DOS of *e_g_* and *t_2g_* spin orbitals for Fe (**b**) and Ni (**c**) under GGA and total DOS (**d**); partial DOS of *e_g_* and *t_2g_* spin orbitals for Fe (**e**) and Ni (**f**) under GGA + *U* (Fe = 5, Ni = 6) schemes.

**Figure 6 materials-12-01844-f006:**
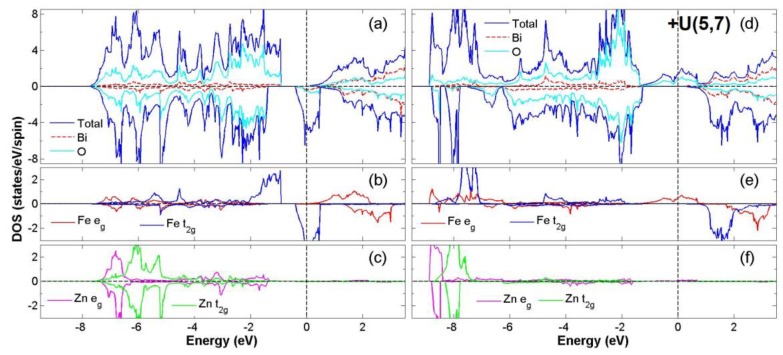
Calculated FiM-Bi_2_FeZnO_6_ total DOS (**a**) and partial DOS of *e_g_* and *t_2g_* spin orbitals for Fe (**b**) and Zn (**c**) under GGA and FM-Bi_2_FeZnO_6_ total DOS (**d**); partial DOS of *e_g_* and *t_2g_* spin orbitals for Fe (**e**) and Zn (**f**) under GGA + *U* (Fe = 5, Zn = 7) schemes.

**Figure 7 materials-12-01844-f007:**
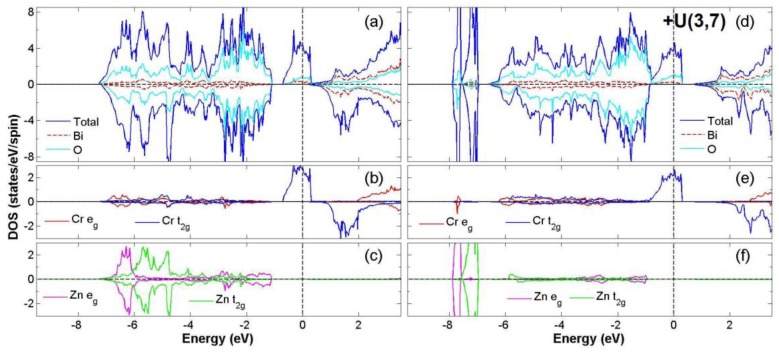
Calculated FiM-Bi_2_CrZnO_6_ total DOS (**a**) and partial DOS of *e_g_* and *t_2g_* spin orbitals for Cr (**b**) and Zn (**c**) under GGA and total DOS (**d**) partial DOS of *e_g_* and *t_2g_* spin orbitals for Cr (**e**) and Zn (**f**) under GGA + *U* (Cr = 3, Zn = 7) schemes.

**Figure 8 materials-12-01844-f008:**
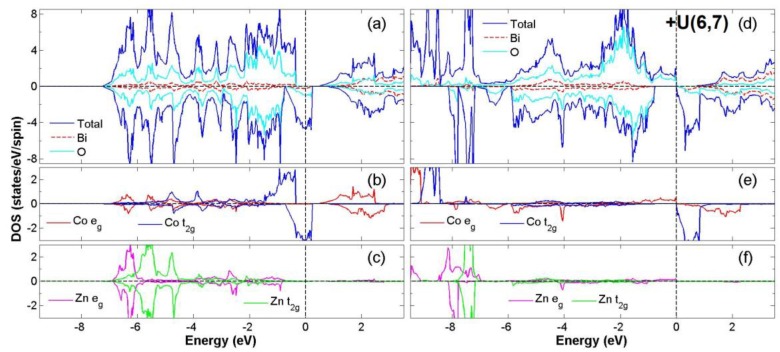
Calculated FiM-Bi_2_CoZnO_6_ total DOS (**a**) and partial DOS of *e_g_* and *t_2g_* spin orbitals for Co (**b**) and Zn (**c**) under GGA and FM-Bi_2_CoZnO_6_ total DOS (**d**) partial DOS of *e_g_* and *t_2g_* spin orbitals for Co (**e**) and Zn (**f**) under GGA+*U* (Co = 6, Zn = 7) schemes.

**Figure 9 materials-12-01844-f009:**
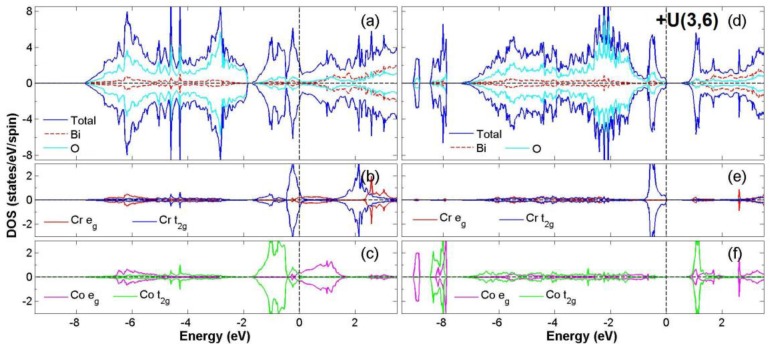
Calculated AF-Bi_2_CrCoO_6_ total DOS (**a**) and partial DOS of *e_g_* and *t_2g_* spin orbitals for Cr (**b**) and Co (**c**) under GGA and total DOS (**d**) partial DOS of *e_g_* and *t_2g_* spin orbitals for Cr (**e**) and Co (**f**) under GGA + *U* (Cr = 3, Co = 6) schemes.

**Figure 10 materials-12-01844-f010:**
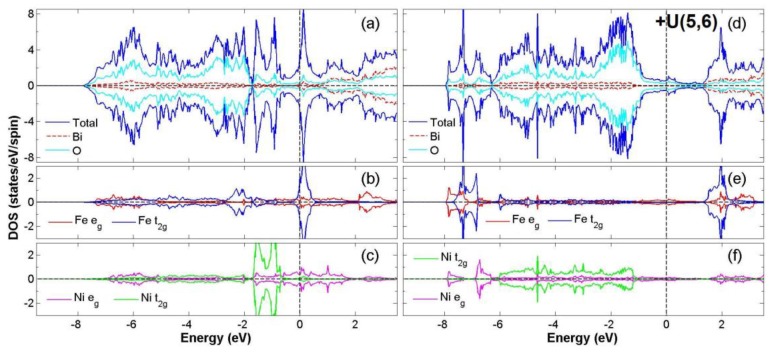
Calculated AF-Bi_2_FeNiO_6_ total DOS (**a**) and partial DOS of *e_g_* and *t_2g_* spin orbitals for Fe (**b**) and Ni (**c**) under GGA and total DOS (**d**) partial DOS of *e_g_* and *t_2g_* spin orbitals for Fe (**e**) and Ni (**f**) under GGA + *U* (Fe = 5, Ni = 6) schemes.

**Figure 11 materials-12-01844-f011:**
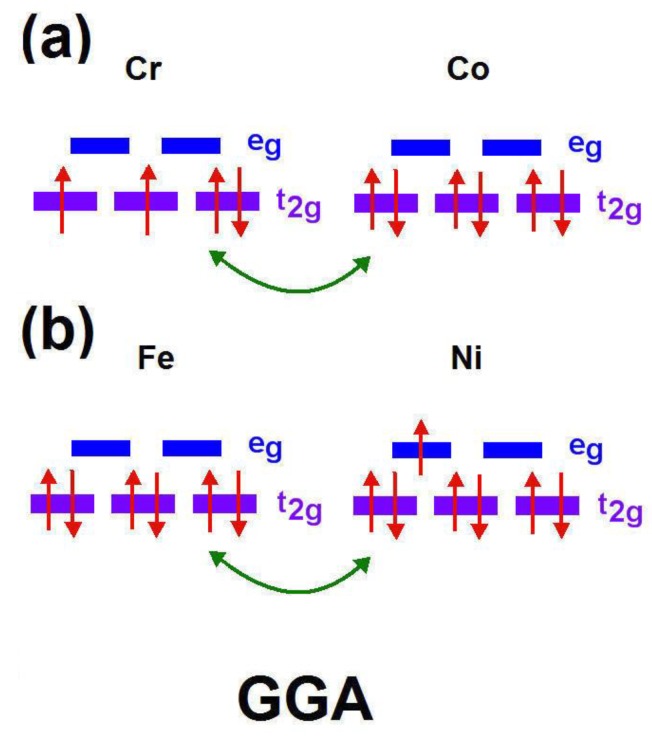
The double exchange interaction configuration for FM-Bi_2_CrCoO_6_ and FM-Bi_2_FeNiO_6_ in the GGA schemes. (**a**) Spin-up electron transfer between Co *d* and Cr *d* in the *t_2_g* via O 2p (**b**) Spin-up electron transfer between Fe *d* and Ni *d* in the *t_2_g* via O 2p.

**Table 1 materials-12-01844-t001:** The final stable structure of tetragonal (I4/mmm, No. 139) and related parameters of the compounds which are fully optimized, where a, c are lattice constants, V_0_ represents the compound volume per f.u. and Bi coordinates (x, y, z) = (0, 0.5, 0.75), *B*(x, y, z) = (0, 0, 0), *B′* (x, y, z) = (0, 0, 0.5), O_1_ (x, y, z) = (0, 0, O_1z_) and O_2_ (x, y, z) = (O_2x_, O_2y_, 0.5) as illustrated in Figure 1.

Bi_2_*BB′*O_6_	a	c/a	V_0_ (Å^3^/f.u.)	O_1z_	O_2x_	O_2y_
CrCo	5.419	1.413	112.461	0.2528	0.2471	0.2471
CrNi	5.453	1.415	114.724	0.2473	0.2526	0.2526
FeNi	5.421	1.413	112.606	0.2453	0.2546	0.2546
FeZn	5.464	1.414	115.332	0.2399	0.2600	0.2600
CrZn	5.505	1.416	118.095	0.2414	0.2587	0.2587
CoZn	5.450	1.415	114.467	0.2400	0.2600	0.2600

**Table 2 materials-12-01844-t002:** Physical properties of the selected FM(FiM)-HM and AF family of Bi_2_*BB*′O_6_ (*B, B′*= *3d* transition metal) with the full structural optimization calculation of GGA and GGA + *U*. In the table below, U_B(B′)_ are the effective parameters used in GGA + *U* calculations for *B*(*B′*). The spin magnetic moments for *B*, *B′*, and the total moment are listed in the table as m_B_, m_B′_, and m_tot_ respectively. Electrons in the spin up and spin down orbitals for *B*(*B′*) elements are listed as well.

Materials Bi_2_*BB*′O_6_	(U*_B_*,U*_B′_*)	Spin Magnetic Moment (μ_B_/f.u.)			*d* Orbital Electrons↑/↓		Band Gap (eV)	∆E (meV/f.u.) FM(FiM)
		m*_B_*	m*_B′_*	m_tot_	*B*	*B′*		
CrCo	(0, 0)	2.573	0.162	3.000	3.465/0.927	3.749/3.575	0.00/0.40	−25
	(3, 6)	2.852	3.071	6.792	3.574/0.804	5.108/2.044	0.00/0.00	903
CrNi	(0, 0)	2.304	1.267	4.000	3.304/1.076	4.767/3.496	0.00/0.65	−97
	(3, 6)	2.303	1.703	4.000	3.272/1.036	5.019/3.320	0.00/1.73	−112
FeNi	(0, 0)	2.146	1.224	4.000	4.143/2.054	4.749/3.528	0.15/0.00	−43
	(5, 6)	3.887	1.644	6.000	4.919/1.094	4.939/3.341	0.00/1.68	34
FeZn	(0, 0)	1.694	−0.022	2.000	3.950/2.269	4.968/4.984	0.97/0.00	−58
	(5, 7)	3.66	0.015	4.000	4.848/1.223	5.022/5.051	0.00/1.52	1125
CrZn	(0, 0)	1.866	−0.013	2.000	3.111/1.274	4.981/4.988	0.00/1.30	−60
	(3, 7)	2.073	−0.008	2.000	3.202/1.165	5.050/5.054	0.00/1.57	−73
CoZn	(0, 0)	0.758	−0.013	1.000	3.995/3.242	4.966/4.976	0.85/0.00	−51
	(6, 7)	3.463	0.105	5.000	5.229/1.782	5.080/5.025	0.65/0.75	99
CrCo(AF)	(0, 0)	2.579	0.057	0.000	3.467/0.925	3.692/3.631	0.00/0.00	–
	(3, 6)	2.799	3.194	0.000	3.561/0.816	5.129/1.952	0.53/0.53	–
FeNi(AF)	(0, 0)	2.805	0.836	0.000	4.439/1.675	4.567/3.735	0.00/0.00	–
	(5, 6)	3.847	1.517	0.000	4.894/1.089	4.912/3.410	0.00/0.0	–

**Table 3 materials-12-01844-t003:** The energy of the final states for Bi_2_*BB*′O_6_ (*B, B′* = *3d* transition metal) with the full structural optimization calculation of GGA and GGA + *U*. In the table below, U_B(B′)_ are the effective parameters used in GGA + *U* calculations for *B*(*B′*).

Materials Bi_2_*BB*′O_6_	(U*_B_*,U*_B′_*)	Final State	E(eV/f.u.)	Materials Bi_2_*BB*′O_6_	(U*_B_*,U*_B′_*)	Final State	E(eV/f.u.) (eV)
CrCo	(0, 0)	AF	−65.880	FeZn	(0, 0)	AF	−57.804
	(3, 6)	AF	−61.827	–	(5, 7)	AF	−56.040
	(0, 0)	FM	−65.905	–	(0, 0)	FiM	−57.862
	(3, 6)	FM	−60.924	–	(5, 7)	FM	−54.915
CrNi	(0, 0)	AF	−63.883	CrZn	(0, 0)	AF	−60.725
	(3, 6)	AF	−60.877	–	(3, 7)	AF	−58.949
	(0, 0)	FM	−63.980	–	(0, 0)	FiM	−60.785
	(3, 6)	FM	−60.989	–	(3, 7)	FiM	−59.022
FeNi	(0, 0)	AF	−61.036	CoZn	(0, 0)	AF	−55.693
	(5, 6)	AF	−57.053	–	(6, 7)	AF	−51.111
	(0, 0)	FM	−61.079	–	(0, 0)	FiM	−55.744
	(5, 6)	FM	−57.019	–	(6, 7)	FM	−51.012

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
