# Peer review of "Half-Metallic Property Induced by Double Exchange Interaction in the Double Perovskite Bi2BB′O6 (B, B′ = 3d Transitional Metal) via First-Principles Calculations"

_materials, 2019, doi:10.3390/ma12111844_

Round 1
Reviewer 1 Report
Half-metallic property induced by double exchange 2 interaction in the double perovskite Bi2BB’O6 (B, B’ = 3d transitional metal) via the First-Principal calculations
Hong-Zong Lin et al
The study is certainly interesting. There are many-fold reasons for this. The materials considered are also interesting because the authors have examined some doped inorganic perovskites, with some hope to discover innovative materials for future technology. There are still some drawbacks of this study, which may be examined in a future investigation. There are some typos, and related other issues, which could be carefully eliminated during revision. I have listed below some points, which the authors would like to answer before I recommend this ms for publication in materials.
1 – The authors have randomly and variably used the terms “GGA+u”, GGA(+u) and “GGA+U”. What is the difference between them? If there is, I ask the authors to clarify them in detail by defining them. If there is none, then I ask the authors to stick to the latter, which is by far common in the DFT-based computational literature.
2 – Why is there a space between between A and B in A B′O3? Is it a typo?
3 - The calculations involved in this ms are based on the density functional theory (DFT) with full-structure optimization by generalized gradient approximation (GGA). What DFT-GGA functional was explicitly used? PBE or PBESol or something else?
4 – Explicit mention of spin states for each state (FM, Anti-FM, etc) is required to be discussed in the materials and methods section.
5 – What was the relative energy difference between the various states of any given system mentioned in 4? This could be clarified in the text in the form of discussion. Why is there a significant difference in DE, with its value changes sign from negative to positive (Table 2)?
6 – The DOS spectra for most of the systems examined are shown in this paper. However, I miss the band structures for the same systems. Does the latter speak the same science, or there is some controversy? I argue this because there is always a significant controversy especially when DFT is used for the exploration of strongly correlated systems involving first row transition metals, viz. VO2, in which, DFT almost fails.
7 – I did not see any discussion concerning the dynamical stability of the materials discussed in this study. Also, I was expecting to see the nature of dielectric behavior, which is missing in this paper. It is important to discuss if this possible. What would the authors say on this?
8 – Whether the perovskite systems examined are organic, inorganic or organic-inorganic, or metal-organic, or something else, is not clear. This must be introduced in the introduction as a separate paragraph. Appropriate references supporting this must be provided.
9 – The conclusion part of the study is not very tightly written. I suggest a revision with some effort.
Author Response
Response to Reviewer 1 Comments
1 – The authors have randomly and variably used the terms “GGA+u”, GGA(+u) and “GGA+U”. What is the difference between them? If there is, I ask the authors to clarify them in detail by defining them. If there is none, then I ask the authors to stick to the latter, which is by far common in the DFT-based computational literature.
Response #1: We are very thankful to the respected reviewer for his useful comments and suggestions. We have accepted the reviewer’s opinion, “GGA+U” and modified these related terms in our revised manuscript.
2 – Why is there a space between between A and B in A B′O3? Is it a typo?
Response #2: We are very thankful to the reviewer. It is a typo and we have checked and modified typo in our revised manuscript
3 - The calculations involved in this ms are based on the density functional theory (DFT) with full-structure optimization by generalized gradient approximation (GGA). What DFT-GGA functional was explicitly used? PBE or PBESol or something else?
Response #3: What we use here is PBE.
4 – Explicit mention of spin states for each state (FM, Anti-FM, etc) is required to be discussed in the materials and methods section.
Response #4: We have combined section 2.2 and 2.3 into as one new section 2.2 in which we have corrected the modified the description for the magnetic states more clearly. We have also added one new Table 2(b) and Figure 9 and 10 for AF state, describing the final energy and magnetic status in detail.
5 – What was the relative energy difference between the various states of any given system mentioned in 4? This could be clarified in the text in the form of discussion. Why is there a significant difference in DE, with its value changes sign from negative to positive (Table 2)?
Response #5: We are very thankful to the reviewer for his comment. The energy levels for final NM states all are large than that of AF and FM(FiM) at least about 1000 meV above, implying that NM cases all here are obviously unstable. This is the reason why we have excluded the list of NM energy state, of course, we have added such status in Table 2(b). For those in the initial state with FM or FiM state, some cases will converge into the state of FiM or FM, while and others will remain in their initial states. So we list all the magnetic status in Table 2(b), including final stable state and the related energy. Applying the GGA+U scheme for Bi2CrCoO6, the PDOS of Cr eg near Fermi level are push to the higher energy level, while the Cr t2g only change a little. Furthermore, the PDOS of Co t2g (spin-up) shifts back to lower energy level, while the PDOS of Co t2g (spin-down) is pushed to the higher energy level, crossing the Fermi energy and breaking the gap. There is a huge variation on the PDOS of Co eg. All the variation of PDOS results in that the total energy value changes sign from negative to positive, finally AF been the stable state as shown in Table2(b) and as Figure 9. When Bi2CrCoO6 is applied with (GGA+U), this compound is an AF-Insulator in which the DE does not exist
6 – The DOS spectra for most of the systems examined are shown in this paper. However, I miss the band structures for the same systems. Does the latter speak the same science, or there is some controversy? I argue this because there is always a significant controversy especially when DFT is used for the exploration of strongly correlated systems involving first row transition metals, viz. VO2, in which, DFT almost fails.
Response #6: We are very thankful to the respected reviewer and for his useful comments and suggestions. This paper mainly focuses on the topic of searching the possible and stable material of FM-HM or FiM-HM, confirming the related energy and supervising its stability. The huge calculations from the various combinations and magnetic states take us roughly one year, finally we are able to find some brilliant results here. For this analysis of their band structures, it is also import and we think that it will be our next paper for considering the time factor. In addition, we have added two figures as shown as Figure 9 and 10 related to AF state. We hope that such more detailed analysis can help us for answering the reviewer's questions. Another question is the calculation related to the cases including V element. We just list some our results for respected reviewer, not shown in the main text. For example, Bi2ScVO6 with GGA+U(2,3), Bi2VFeO6 with GGA+U(3,5) and Bi2VNiO6 with GGA+U(3,6) are all finally as stable FM-metal, while Bi2VZnO6 with GGA+U(3,7) is a stable AFM-metal. So, that is the reason why we exclude the cases with V element. In short, in view of calculations we decide not to discuss V-related cases here. We are very thankful to the reviewer and his useful comments and suggestions again.
7 – I did not see any discussion concerning the dynamical stability of the materials discussed in this study. Also, I was expecting to see the nature of dielectric behavior, which is missing in this paper. It is important to discuss if this possible. What would the authors say on this?
Response #7: It is also a big issue and important for the dynamical stability or for the research of dielectric behavior. However, this paper mainly focuses on the topic of looking for the stable HM material in which we try to find the final energy level and to confirm its stability for the various combinations and magnetic states, as well as the GGA+U schemes. This issue for the dynamical stability or the dielectric behavior will be our next paper's topic. We are very thankful to the reviewer and his useful comments and suggestions again.
8 – Whether the perovskite systems examined are organic, inorganic or organic-inorganic, or metal-organic, or something else, is not clear. This must be introduced in the introduction as a separate paragraph. Appropriate references supporting this must be provided.
Response #8: We accept the respected reviewer's opinion and added one paragraph to describe the history related the researches of organic, inorganic or organic-inorganic, or metal-organic issues. For this viewpoint of this paper, we mainly focuses on the topic of searching HM material from the inorganic double perovskite compounds, like inorganic Sr2FeMoO6. We have also added some important references here.
9 – The conclusion part of the study is not very tightly written. I suggest a revision with some effort.
Response #9: We have checked and modified this conclusion in our revised manuscript. The new added Table 2(b), modified Table 2(a), Figures 9 and 10 are also included and solidified. These convincing evidences are added into this manuscript and conclusion. We are very thankful to the respected reviewer again.
Reviewer 2 Report
Using ab-inito density functional theory calculations, this work explores candidates for possible Bi-based double perovskite oxides that can show half-metallic and ferromagnetic / ferrimagnetic phase. For future spintronic applications, it would be nice to have multiple possible candidates that show stable ferromagnetism and good metallicity, so this work may found to be meaningful. Indeed, the calculations are, in my opinion, done relatively well, so the results presented in this manuscript can be a useful reference for further in-depth studies in the future. Hence I would like to recommend this work for the publication in this journal, but after some improvements.
Here the double exchange mechanism is suggested to be the main mechanism to make the candidate systems to be half-metallic, but the mechanism itself is very vaguely stated, like "The double exchange is a type of electrons (spin-up or spin-down) change that may arise between ions in different oxidation state. This theory can predict the relative ease with which an electron may be exchanged between two species, ..." I would like to suggest the authors to clarify more about what the double exchange mechanism. Also, the above statement appears again exactly same in the beginning of Sec. 3.3, which looks like a copy-and-paste and does not look professional at all.
A related issue is the following; some of the candidate materials contain Zn at the B-sites, which is in a d^10 configuration and is nonmagnetic. And double exchange is a mechanism happening between two magnetically active (i.e. partially-filled d-shell) with different charge status. So, double exchange is irrelevant in compounds with Zn, but the authors are still discussing double exchange in those compounds as well, and it does not look appropriate at all.
For example, at the end of Section 3.2, it was stated that "For Bi2CoZnO6, this compound is very similar to Bi2FeZnO6 in many ways. The double exchange interaction of Co_3d-O_2p-Zn_3d orbitals causes the hybridization in the energy of -8.5eV to -1.0 eV for spin up channel and 0 eV to 1.0 eV for spin down channel." Simply speaking, since Zn d-shell is fully filled (d^10) it is not even possible for the electrons from Co to hop to Zn site (see the absence of Zn states in PDOS in Fig. 7) so no double exchange exist. Also, it is very dubious that "double exchange causes the hybridization in the energy of -8.5 to -1.0eV"; it is NOT that the double exchange causes the hybridization, but it exists on top of the d-p hybridized band structure. The energy scale of magnetic exchange interactions is about 0.1 eV at most in usual transition metal compounds, which is order-or-magnitudes smaller than the d-p hybridization energy scale.
These above problematic statements seriously undermines the scientific credibility of this manuscript. I strongly suggest the authors to go back to the previous literatures, understand the double exchange mechanisms, and state it correctly in the manuscript. Also interpretations of several DOS plots should be changed as well.
Another minor issue is, there are several expressions that was copy-and-pasted with just different compound name, like "For XXXXX, this compound is very similar to YYYYY in many ways. The double exchange interaction of AA_3d-O_2p-BB_3d orbitals causes the hybridization in the energy of -*.*eV to -*.* eV for spin up channel and 0 eV to 1.0 eV for spin down channel." I understand that several results are quite similar so it is very tempting to use the exactly the same expressions, but for readers it looks very funny and unprofessional.
Finally, please use an english editing/correction service before resubmitting. Some paragraphs were very hard to comprehend (for example, Section 2.2).
Author Response
Response to Reviewer 2 Comments
1 –Here the double exchange mechanism is suggested to be the main mechanism to make the candidate systems to be half-metallic, but the mechanism itself is very vaguely stated, like "The double exchange is a type of electrons (spin-up or spin-down) change that may arise between ions in different oxidation state. This theory can predict the relative ease with which an electron may be exchanged between two species, ..." I would like to suggest the authors to clarify more about what the double exchange mechanism. Also, the above statement appears again exactly same in the beginning of Sec. 3.3, which looks like a copy-and-paste and does not look professional at all.
Response #1: We are very thankful to the respected reviewer and for his useful comments and suggestions. We have added one paragraph to introduce the history of double exchange. The related references as ref [8,9, 17, 31-36] and as the book for the DE issue, ref 34 Quantum theory of magnetism, are included in this manuscript. We have also re-plotted the Figure 11 and seriously analyze the pattern for some DOS Figures. The DE effect related to the Zn will be corrected because of the status of full-occupied d orbitals and almost zero magnetic moment. We have also rewritten and modified the main text in Sec. 3.3.
2 –A related issue is the following; some of the candidate materials contain Zn at the B-sites, which is in a d^10 configuration and is nonmagnetic. And double exchange is a mechanism happening between two magnetically active (i.e. partially-filled d-shell) with different charge status. So, double exchange is irrelevant in compounds with Zn, but the authors are still discussing double exchange in those compounds as well, and it does not look appropriate at all.
Response #2: We are very thankful to the respected reviewer. Indeed, the DE effect, considered as the effect of the related characteristic scalar product (Si · Sj ) via O 2p, in this manuscript has been rewritten here and some cases related to Zn element will be corrected and deleted in this manuscript because of full-occupied d orbitals and the small energy scale of magnetic exchange interactions. Figure 11, DE effect, and has been modified and described only two compounds of Bi2CrCoO6 and Bi2FeNiO6 for analysis. When applying the GGA+U scheme for Bi2CrCoO6, the PDOS of Cr eg near Fermi level are push to the higher energy level, while the Cr t2g only change a little. Furthermore, the PDOS of Co t2g (spin-up) shifts back to lower energy level, while the PDOS of Co t2g (spin-down) is pushed to the higher energy level, crossing the Fermi energy and breaking the gap. There is a huge variation on the PDOS of Co eg. All the variation of PDOS results in that the total energy value changes sign from negative to positive, finally AF been the stable state as shown in Table 2(a), 2(b) and Figure 9. When Bi2CrCoO6 is applied with (GGA+U), this compound is an AF-Insulator in which the DE does not exist. The same situation is also like Bi2FeNiO6 applied with (GGA+U), while the final stable state is an AF-metal.
3 –For example, at the end of Section 3.2, it was stated that "For Bi2CoZnO6, this compound is very similar to Bi2FeZnO6 in many ways. The double exchange interaction of Co_3d-O_2p-Zn_3d orbitals causes the hybridization in the energy of -8.5eV to -1.0 eV for spin up channel and 0 eV to 1.0 eV for spin down channel." Simply speaking, since Zn d-shell is fully filled (d^10) it is not even possible for the electrons from Co to hop to Zn site (see the absence of Zn states in PDOS in Fig. 7) so no double exchange exist. Also, it is very dubious that "double exchange causes the hybridization in the energy of -8.5 to -1.0eV"; it is NOT that the double exchange causes the hybridization, but it exists on top of the d-p hybridized band structure. The energy scale of magnetic exchange interactions is about 0.1 eV at most in usual transition metal compounds, which is order-or-magnitudes smaller than the d-p hybridization energy scale.
Response #3: We are very thankful to the respected reviewer again. The DE effect related to the Zn element in this manuscript have been corrected. We accepted reviewer’s opinion that DE exists on top of the d-p hybridized band structure. Therefore, we have also modified and fixed the viewpoint in Figure 11 and checked the distribution of PDOS from Figures 3-8. The detailed analysis for PDOS are modified and also embedded into the related sections and paragraphs in this manuscript.
4 –These above problematic statements seriously undermines the scientific credibility of this manuscript. I strongly suggest the authors to go back to the previous literatures, understand the double exchange mechanisms, and state it correctly in the manuscript. Also interpretations of several DOS plots should be changed as well.
Response #4: We have accepted the respected reviewer’s opinion and have studied some papers and books in ref [8,9, 17, 31-36], especially for chapter 5 of ref 34 Quantum theory of magnetism. We have also re-plotted the Figure 11 and seriously state the pattern for some DOS Figures. The discussion of DE effect and Zn element will be corrected because of full-occupied d orbitals. Figure 11 have been modified and checked again.
5 –Another minor issue is, there are several expressions that was copy-and-pasted with just different compound name, like "For XXXXX, this compound is very similar to YYYYY in many ways. The double exchange interaction of AA_3d-O_2p-BB_3d orbitals causes the hybridization in the energy of -*.*eV to -*.* eV for spin up channel and 0 eV to 1.0 eV for spin down channel." I understand that several results are quite similar so it is very tempting to use the exactly the same expressions, but for readers it looks very funny and unprofessional.
Response #5: According to the similar results in the main text, we accept the reviewer’s opinion and have checked and modified these expressions in this revised manuscript.
6 –Finally, please use an english editing/correction service before resubmitting. Some paragraphs were very hard to comprehend (for example, Section 2.2).
Response #6: We have combined section 2.2 and 2.3 into one part of section 2.2 and rewritten some paragraphs. We have also checked and corrected all the manuscript and try to make a clearly statement. We also let an English expert to revise our article. We are very thankful to the respected reviewer.